# Circulating isomiRs May Be Superior Biomarkers Compared to Their Corresponding miRNAs: A Pilot Biomarker Study of Using isomiR-Ome to Detect Coronary Calcium-Based Cardiovascular Risk in Patients with NAFLD

**DOI:** 10.3390/ijms25020890

**Published:** 2024-01-10

**Authors:** Nataly Makarenkov, Uri Yoel, Yulia Haim, Yair Pincu, Nikhil S. Bhandarkar, Aryeh Shalev, Ilan Shelef, Idit F. Liberty, Gal Ben-Arie, David Yardeni, Assaf Rudich, Ohad Etzion, Isana Veksler-Lublinsky

**Affiliations:** 1Department of Clinical Biochemistry and Pharmacology, Faculty of Health Sciences, Ben-Gurion University of the Negev, Beer-Sheva 84105, Israel; nataly@post.bgu.ac.il (N.M.); uriy@bgu.ac.il (U.Y.); nsbhandarkar08@gmail.com (N.S.B.); 2Department of Software & Information Systems Engineering, Faculty of Engineering, Ben-Gurion University of the Negev, Beer-Sheva 84105, Israel; vaksler@post.bgu.ac.il; 3The Endocrinology Unit, Soroka University Medical Center, Beer-Sheva 84101, Israel; 4Cardiology Department, Soroka University Medical Center, Beer-Sheva 84101, Israel; 5Department of Diagnostic Imaging, Soroka University Medical Center, Beer-Sheva 84101, Israel; 6Diabetes Clinic, Soroka University Medical Center, Beer-Sheva 84101, Israel; iliberty@bgu.ac.il; 7Department of Gastroenterology and Liver Diseases, Soroka University Medical Center, Beer-Sheva 84101, Israelohadet@clalit.org.il (O.E.)

**Keywords:** NAFLD, coronary artery calcium (CAC) score, cardiovascular risk, microRNA, isomiR (iso-microRNA), biomarker

## Abstract

Circulating miRNAs are increasingly being considered as biomarkers in various medical contexts, but the value of analyzing isomiRs (isoforms of canonical miRNA sequences) has not frequently been assessed. Here we hypothesize that an in-depth analysis of the full circulating miRNA landscape could identify specific isomiRs that are stronger biomarkers, compared to their corresponding miRNA, for identifying increased CV risk in patients with non-alcoholic fatty liver disease (NAFLD)—a clinical unmet need. Plasma miRNAs were sequenced with next-generation sequencing (NGS). Liver fat content was measured with magnetic-resonance spectrometry (MRS); CV risk was determined, beyond using traditional biomarkers, by a CT-based measurement of coronary artery calcium (CAC) score and the calculation of a CAC score-based CV-risk percentile (CAC-CV%). This pilot study included n = 13 patients, age > 45 years, with an MRS-measured liver fat content of ≥5% (*wt*/*wt*), and free of overt CVD. NGS identified 1103 miRNAs and 404,022 different isomiRs, of which 280 (25%) and 1418 (0.35%), respectively, passed an abundance threshold. Eighteen (sixteen/two) circulating miRNAs correlated positively/negatively, respectively, with CAC-CV%, nine of which also significantly discriminated between high/low CV risk through ROC-AUC analysis. IsomiR-ome analyses uncovered 67 isomiRs highly correlated (R ≥ 0.55) with CAC-CV%. Specific isomiRs of miRNAs 101-3p, 144-3p, 421, and 484 exhibited stronger associations with CAC-CV% compared to their corresponding miRNA. Additionally, while miRNAs 140-3p, 223-3p, 30e-5p, and 342-3p did not correlate with CAC-CV%, specific isomiRs with altered seed sequences exhibited a strong correlation with coronary atherosclerosis burden. Their predicted isomiRs-specific targets were uniquely enriched (compared to their canonical miRNA sequence) in CV Disease (CVD)-related pathways. Two of the isomiRs exhibited discriminative ROC-AUC, and another two showed a correlation with reverse cholesterol transport from cholesterol-loaded macrophages to ApoB-depleted plasma. In summary, we propose a pipeline for exploring circulating isomiR-ome as an approach to uncover novel and strong CVD biomarkers.

## 1. Introduction

MicroRNAs (miRNA) are short (~20–24 nucleotides) non-coding RNA sequences. They function as post-transcriptional, negative regulators of gene expression by destabilizing coding mRNA transcripts, and/or by interfering with the translation of mRNA to protein [1]. Although this function depends on their intracellular actions, miRNAs are also found in the circulating blood, carried either free, within extracellular vesicles (exosomes), and/or by other particles like lipoproteins (particularly HDL) [2,3,4]. Connecting circulating miRNAs to their cellular origin is not straightforward, but it is largely assumed that the dysregulation of extracellular miRNAs is reflective of changes in their intracellular abundance [5], rendering circulating miRNAs attractive biomarkers for various disease states [6,7,8] in addition to their potential as therapeutic agents [9].

It is noteworthy that not all miRNA species are detectable when using probe-hybridization–based methods to profile the miRNA “landscape” [10]. In particular, these techniques fail to detect IsomiRs, or variances in miRNAs’ length and/or substitutions of single nucleotide within the specific miRNA sequence [11]. The detection of isomiRs became possible with the advent of next-generation sequencing (NGS) methodologies. In addition to the “canonical” miRNA sequence (the sequence that is reported in databases), NGS captures miRNA isoforms that can largely be classified into 5′ or 3′ isomiRs, with changes in length at the 5′ or 3′ end, respectively; polymorphic-isomiRs, with identical lengths but with nucleotide changes within the mature sequence; and mixed-type-isomiRs, with changes in length and sequence. IsomiRs with length variations could result from imprecise cleavage by miRNA biogenesis enzymes Drosha and Dicer, or by exonuclease nibbling activity at the ends of the (mature) miRNA. In both cases, the resulting isomiRs are “templated isomiRs”, as they match the reference genomic sequence. Variants with length differences could also arise from the post-transcriptional addition of one or more bases by Nucleotidyl Transferases. These variants are “non-templated isomiRs” since their ends may not match the parent gene. Polymorphic isomiRs are less frequent, and they contain internal nucleotides which are different from the genomic sequence due to miRNA editing events [11]. Importantly, isomiRs not only expand the miRNA repertoire, but they may also introduce additional tissue- and disease-related specificities [12]. Mechanistically, even small variations in length (5′-isomiRs) and/or sequence could modify the seed sequence—nucleotides at positions 2–7 that determine the gene target repertoire of the miRNA. Though 3′ isomiRs have the same seed sequence as the canonical miRNA, they may have altered miRNA loading into the miRNA-induced silencing complex (miRISC), miRNA stability, and miRNA targeting characteristics compared to the canonical form [13,14,15,16,17]. This implies that isomiRs could have distinct or divergent functions compared to their canonical form, thereby uniquely participating in the regulation of disease pathways. 

Micro-RNAs have been extensively studied in the context of non-alcoholic fatty liver disease (NAFLD, now also termed metabolic dysfunction-associated fatty liver disease, MAFLD); a condition that describes a spectrum of diseases ranging from relatively benign fat accumulation in hepatocytes (i.e., simple steatosis), to fatty infiltration accompanied by varying degrees of inflammation (i.e., steatohepatitis) and liver fibrosis. The latter may eventually progress to cirrhosis, thereby constituting a major risk factor for developing hepatocellular carcinoma [18]. The prevalence of NAFLD exceeds 25% of the global adult population [19], with even higher rates among people with obesity and diabetes [20]. Although posing a significant risk for mortality attributed to liver disease, studies suggest that the main public health impact and health risk for patients with NAFLD is an elevated risk of cardiovascular (CV) morbidity and mortality. Indeed, NAFLD is an independent risk factor for the occurrence of CV events in addition to traditional precipitators such as hypertension, diabetes, and dyslipidemia [21,22,23]. Furthermore, NAFLD is associated with surrogate markers of subclinical atherosclerosis, such as an elevated coronary artery calcium (CAC) score [24] and increased carotid intimal thickness [25]. Currently, a major unmet clinical challenge is in distinguishing patients with NAFLD who will follow a relatively benign clinical course with a CV risk that can be well-estimated by classical risk factors, from those in whom NAFLD poses a particularly elevated risk of developing CV diseases (CVD). 

In this proof-of-concept pilot study, we hypothesize that an NGS-based assessment of the isomiR landscape (the “isomiR-ome”) can uncover specific circulating miRNAs that particularly correspond to the level of CV risk in carefully characterized patients with NAFLD but without overt cardiovascular diseases: An excessive (≥5% *w*/*w*) liver TG content was confirmed with magnetic resonance spectrometry (MRS), and CV risk was assessed using traditional risk factors and by measuring the coronary artery calcium (CAC) score. Our findings demonstrate that specific circulating isomiRs constitute putative superior biomarkers compared to their corresponding miRNA, and, specifically herein, may address the unmet clinical need for improved tools to identify NAFLD patients with an excessively high CV risk.

## 2. Results

We developed a pipeline for identifying specific isomiRs that, compared to the corresponding (total) miRNA counts (i.e., all isoforms of each miRNA), can be more robustly, or even uniquely, connected with disease risk. We exemplify that our pipeline can identify putative biomarkers for increased cardiovascular risk in a small but well-characterized group of patients with non-alcoholic fatty liver disease (Figure 1). 

Our pipeline starts with collecting blood samples and assessing clinical parameters and scores from a relevant population. Then, blood samples are checked for hemolysis, and samples that pass the recommended thresholds are subjected to miRNA sequencing with NGS. miRNA libraries are then checked for the miRNAs’ detection rates, and outlier libraries are removed. Then, a bioinformatic analysis is carried out to compute any correlations between a clinical score and (1) the abundance of circulating miRNAs, using the traditional approach in which counts of all isoforms contribute to the total miRNA count, and (2) the abundance of specific circulating isomiRs. For miRNA/isomiRs that exhibit a correlation above a certain threshold we (1) compute the *p*-values by using a permutation test and further adjust them through common methods, and (2) assess their discriminative power through ROC-AUC analysis by stratifying the patients into groups based on traditional cutoffs for the clinical score. miRNAs and isomiRs that exhibit higher correlations than their corresponding miRNAs are then subjected to targetome analysis. Finally, further biological plausibility of our findings is obtained by correlating circulating miRNA/isomiR abundance and reverse cholesterol transport activity assessed with Apo-B-depleted plasma. 

### 2.1. Patients and Clinical Parameters

Thirteen patients, nine males and four females, who were ≥45 years old (median: 55; range: 46–79; in 12/13 of the patients the age range was 46–66 years), with an MRS-confirmed fatty liver (liver fat ≥ 5% *wt*/*wt*) were recruited to the study. Baseline individual characteristics are shown in Appendix A, and data for the entire cohort are shown in Table 1. Notably, based on the CAC-CV% calculation, which was used as a reference standard, six participants had a high-risk score for CVD (CAC-CV% in the ≥ 75th percentile), while seven were in the range of the 0–66th percentiles. CV-risk scores obtained with three different calculators (Table 1) were significantly inter-correlated but did not correlate with the CAC-CV% (Appendix A). When assessing their ability to discriminate high-risk from low-risk CAC-CV% using ROC analysis, only the ACC/AHA calculator had a mild, statistically significant discriminative power (ROC-AUC = 0.81, 95%CI: 0.54–1). This partially reflects the fact that calculators like the Framingham CV risk calculator are not designated for NAFLD patients, thereby demonstrating the unmet clinical need for biomarkers to identify NAFLD patients with excessive CV risk. 

### 2.2. Sequencing of Circulating miRNAs

Circulating miRNAs were sent for NGS after stringent quality control to ensure a low contribution of red blood cell lysis (real-time PCR measurements of miR-451a and 23a-3p), and comparable recovery during isolation (Cel-miR-39 spike-in procedure), as detailed in the Methods section. Individual sample quality control results are shown in Appendix A, demonstrating that sample number five exhibited a relatively low number of unique miRNA sequences. The intercorrelation between the abundance of miRNA sequences obtained from the 13 libraries showed a particularly low correlation between sample five and the other samples (Appendix A) (other libraries inter-correlated with a correlation coefficient ≥ 0.6). Moreover, when assessing the number of common miRNA sequences in at least three or five of the samples, their number in sample five was particularly low, which was statistically identified as an outlier (Appendix A). Hence, this sample had to be excluded from further analyses.

### 2.3. Correlation between miRNAs and Coronary Atherosclerosis (CAC-CV%)

We first considered circulating miRNAs that were sufficiently abundant, i.e., whose average count was above 5 reads per million (RPM), and the maximal count among the samples was above 50 RPM. In this analysis, the miRNA count includes the number of canonical sequences and all their related variant isomiRs. Out of 1103 identified miRNA sequences, 280 passed these abundance criteria and were further correlated with patients’ CAC-CV% using Spearman’s correlation. We used a cutoff Spearman’s coefficient of (ρ) ≥ 0.55, and assessed statistical significance using a permutation approach followed by multiple testing correction, as detailed in the Methods section. Eighteen miRNAs achieved *p*-adj < 0.05 and were considered as CAC-CV%-correlated miRNAs (Table 2, Figure 2), many of which were inter-correlated (Appendix A). Nine of the miRNAs also significantly discriminated between high/low CV risk by ROC-AUC analysis (Table 2, Figure 2). Sixteen exhibited a positive correlation and two exhibited a negative correlation with CAC-CV%. The dynamic range of their abundance, i.e., the ratio between the highest and lowest RPM, was 2.5–131-fold. Additionally, this set of miRNAs already had known associations with CV-related diseases and pathways: using the Mammalian ncRNA-Disease Repository (MNDR) feature of the miEAA online tool [27], 15 of the miRNAs were enriched in at least one CVD-related category, including vascular disease (fifteen miRNAs), atherosclerosis (seven miRNAs), coronary artery disease (six miRNAs), and vascular calcification (five miRNAs) (Figure 3A). Furthermore, many of these miRNAs’ validated targets were enriched in the functional KEGG pathways directly related to CVD, such as ‘fluid shear stress and atherosclerosis’ and ‘cardiac muscle contraction’, or functional pathways highly related to the pathogenesis of CVD, including the regulation of the ‘insulin signaling pathway’ and ‘insulin resistance’ (17 and 16 miRNAs, respectively), ‘cholesterol metabolism’ and ‘ABC transporters’ (12 and 11 miRNAs, respectively), and fat metabolism (Figure 3B). Jointly, in this cohort of patients with NAFLD, 18 circulating miRNAs, each assessed as the total abundance of its canonical and isomiR forms, correlated with patients’ coronary artery atherosclerosis load. 

### 2.4. Expanding the Circulating miRNA Repertoire by Considering isomiRs

In this study, we hypothesized that exploring isomiRs can uncover putative circulating biomarkers for CV risk beyond those obtained when considering the corresponding, total (including all sequence variations) miRNAs. To address our hypothesis, we took advantage of the NGS approach, which is used to assess the full repertoire of circulating miRNAs. Applying the same pipeline used for the miRNAs (Figure 2), out of >400,000 unique isomiR sequences, we identified 67 isomiRs that correlated with CV risk in our patients. Twenty-one of the isomiRs belong to CAC-CV%-correlated miRNAs. Intriguingly, seven of them exhibited a stronger correlation with CAC-CV%, a lower *p*-value, and a higher dynamic range of expression compared to the total specific miRNA count (Figure 4A). Five of these isomiRs showed discriminative power. Moreover, one and three isomiRs of miRNA 144-3p and 101-3p, respectively, could discriminate between high/low CV risk (based on a CAC-CV% cutoff of 75%), although the ROC-AUC value for the corresponding (total count) miRNA was non-discriminative (Figure 4A, Table 2 and Appendix A). An additional 46 isomiRs exhibited a Spearman’s coefficient of correlation above 0.55, although their corresponding total miRNA count did not significantly correlate with CAC-CV%. Of these isomiRs, nine had a seed-modified sequence (Figure 4B). As mentioned in the Introduction, specific isomiRs can not only reflect changes in their biogenesis and tissue of origin but may also alter the mRNAs targeted by these miRNAs. For each of these seed-changing isomiRs and their respective canonical form, we assessed the “isomiR-specific targetome” using the miRmut2go webtool [28], since it may reflect the possible pathways that uniquely and mechanistically link the abundance of these isomiRs with elevated/accelerated processes that are reflected in increased CV risk. Then, each targetome was searched for overrepresented pathways using the DAVID webtool [29], focusing on KEGG pathways and Wikipaths. IsomiR-specific predicted gene targets for four of the isomiRs—miR-140-3p, miR-223-3p, miR-30e-5p, and miR-342-3p—exhibited enrichment in functional pathways related to CVD, which were not apparent in the targetome of their canonical miRNA sequence (Figure 5). These included insulin signaling and/or insulin resistance for all four isomiRs, circadian rhythm regulation (isomiRs 140-3p, 223-3p, and 342-3p), and thermogenesis (isomiR 140-3p), providing biological plausibility for the specific isomiR’s, but not its canonical miRNA sequence, association with CVD. Furthermore, ROC-AUC analysis revealed that the AUC of all four isomiRs is greater than the AUC of the corresponding total miRNA, and two of them significantly (ROC-AUC > 0.8) discriminated between high/low CV risk (Figure 6A–D).

Finally, we assessed whether the association between isomiRs and CV risk may be mediated by HDL function and the process of reverse cholesterol transport (RCT) using Apo-B-depleted plasma as a cholesterol acceptor from cultured, cholesterol pre-loaded macrophages. RCT was inversely correlated with CAC-CV% (r = −0.577, *p* = 0.049) (Figure 6E), and mean RCT was higher for the lower CAC-CV% (based on a CAC-CV% cutoff of 75%); however, this difference was not statistically significant (*p*-value = 0.09). Interestingly, RCT was not correlated with total HDL (r = 0.467, *p* = 0.126) and total HDL was not correlated with CAC-CV% (r = −0.124, *p*-value = 0.702). Yet importantly, two of the seed-changing CAC-CV%-correlated isomiRs showed a significant correlation with RCT (negative for isomiR-140-3p and positive for isomiR-23-3p), while no such correlation was observed with their corresponding total miRNA (Figure 6F,G). 

## 3. Discussion

Circulating miRNAs are increasingly being studied as potential biomarkers for predicting disease risk and staging. Frequently, however, results remain inconclusive due to the overall poor reproducibility between studies, cohorts, and populations. There are likely multiple reasons for this variability, including methodological ones. Particularly, i. only considering a pre-selected set of candidate miRNA biomarkers rather than using an unbiased approach to measure the full repertoire of circulating miRNAs. ii. The use of hybridization-based higher-throughput technologies (microarrays, Nanostring©), which allow us to expand the number of miRNAs being considered and employ an unbiased approach, but cannot discriminate between the different miRNA isoforms and a canonical miRNA sequence. Mapping the entire miRNA repertoire, including all isomiRs, requires NGS. When isomiRs were initially noticed, they were suspected to reflect technical errors introduced by NGS. Yet, it is now well established that isomiRs do reflect the biological variability of miRNAs which results from alterations in miRNA biogenesis and/or post-transcriptional modifications [11]. IsomiRs vary in length and/or in the nucleotide composition compared to their canonical miRNA sequence. Since the miRNA seed sequence is a major (even if not the sole) determinant of miRNA–gene target interaction, isomiRs with modified seed sequences exhibit an altered targetome compared to the “parent”/canonical miRNA, and hence, may vary in their biological activity. The biological relevance of isomiRs was proposed by demonstrating tissue and disease-specific preference of certain isomiRs [12,30,31,32,33]. Hence, when studying miRNAs’ biological roles and/or whether they can constitute biomarker, NGS-enabled isomiR assessment addresses many of the shortcomings of other approaches. Our study proposes a rational pipeline for the identification of specific isomiRs that, compared to the total counts (i.e., all isoforms) of a specific miRNA, can be more robustly, or even uniquely, connected with disease risk.

In this study, we propose and exemplify how such a pipeline can identify biomarkers for a clinically unmet need. Our pipeline incorporates rigorous quality controls prior to and following NGS, which include assessment of hemolysis and miRNA detection rates across samples, respectively. Importantly, this pipeline addresses several “generic challenges” that biomarker discovery studies may face. First, we identify putative-biomarker circulating miRNAs through correlation tests, thereby minimizing the risk of misclassification bias that characterizes the stratification of patients into groups based on clinical score cutoffs. Second, as hundreds of miRNAs (and many thousands of isomiRs) are considered to be potential biomarkers, we address multiple comparison-related biases using a permutation-test. Only then we proceed to create categorical definitions and compute the discriminative power of the potential biomarkers using ROC-AUC. 

Indeed, the robustness of this pipeline could be demonstrated in this pilot proof-of-principle study: Despite a very limited number of participants, we identified 18 circulating miRNAs that strongly correlated with CAC-CV% (Spearman’s correlation coefficient ≥ 0.55). Interestingly, enrichment analysis of these miRNAs revealed pathways such as insulin signaling, glycolysis, and TNF signaling, which are central to NAFLD and/or NASH pathogenesis. When considering the entire isomiR-ome using the same filtering scheme for significant abundance and strength of association, we identified specific isomiRs for some of these 18 miRNAs that exhibited a stronger association with CAC-CV% than their respective total abundance in the circulation. More importantly, we identified isomiRs that exhibited a significant correlation with coronary atherosclerosis burden, although their total miRNA abundance did not correlate with CAC-CV%. Some of these were seed-changing isomiRs, with a potentially unique expected miRNA targetome. Indeed, predicted targets of these CAC-CV%-correlating isomiRs are enriched in biological pathways which are highly relevant to the pathogenesis of atherosclerosis in NAFLD; pathways which, amongst the targets of the respective canonical miRNA sequence, were not enriched. These provide a potential biological plausibility underlying the association between such isomiRs and coronary atherosclerosis and demonstrate the potential benefit of investigating the full circulating miRNA repertoire when attempting to identify potentially clinically useful miRNA biomarkers of subclinical CV disease. Moreover, our results suggest that decreased RCT activity in plasma is a potential mechanism by which specific isomiRs may impact CV risk and provide further support for the biological plausibility of our statistical and bioinformatics-based findings. However, as our sample is small in this pilot study, the mechanism/s that connect miRNA with RCT, and other potential pathways related to atherosclerosis, should be further investigated.

Multiple studies have attempted to identify circulating miRNAs that would be helpful in assessing the severity of NAFLD, particularly for discriminating simple steatosis (a relatively benign condition) from steatohepatitis that has a much higher risk of subsequently developing into liver fibrosis and cirrhosis [34,35]. In parallel, miRNAs which are indicative of CVD severity, have also been proposed [8,36]. In our small but well-characterized cohort of patients with NAFLD, three common CV-risk calculators based on clinical parameters did not correlate with CAC-CV%, a well-established CV risk assessment tool [26]. This highlights the possible advancement that could be offered by more sensitive biomarkers for undiagnosed CVD, a major cause of morbidity and mortality in these patients [21]. Clearly, this study does not have the power to determine which specific miRNAs and/or isomiRs may serve as biomarkers for sub-clinical CVD in patients with NAFLD. Furthermore, this study only evaluated subclinical, CVD-associated, large coronary vessels, while it has recently been acknowledged that the determinants of coronary-event risk have a stronger association with small rather than large coronary vessel disease [37,38]. Nevertheless, the results of the present study are important as they suggest that an in-depth assessment of the circulating isomiR-ome, i.e., the full repertoire of circulating miRNAs and their isoforms, may uncover novel biomarkers that can be utilized for the improved stratification of NAFLD patients. Moreover, bioinformatic analysis and RCT experiments suggested pathways that may shed light on the mechanisms of action of miRNAs/isomiRs in target tissues. Validating these findings in a larger cohort will support the development of isomiR-based diagnostics as a non-invasive tool for the assessment of subclinical CVD in NAFLD, and could highlight circulating miRNAs and isomiRs that should be mechanistically investigated to test their likelihood of acting as mediators in the pathogenesis of CVD in NAFLD patients.

Our study has noteworthy limitations and strengths. Given that there are still barriers to the routine use of NGS, our cohort is small, limiting our ability to utilize big-data methodologies to propose an isomiR-based prediction algorithm for CAC-CV% risk. Clearly, more research is required to uncover the roles and utility of isomiRs in NAFLD-related CV risk assessment and mechanisms. However, patients are well characterized by MRS and CAC, providing quantitative assessment of hepatic fat content and coronary calcification, respectively. These demonstrate that our cohort encompasses the full range of CV risk among patients with NAFLD. Despite the limited number of patients, we did find a correlation between numerous miRNAs/isomiRs and CAC-CV%. This suggests that miRNA and isomiRs identified by NGS techniques may be utilized, following a validation study, to improve CV-risk stratification among patients with NAFLD. The study’s strengths include stringent quality control of samples and an in-depth bioinformatic analyses of the miRNA sequences, enabling the consideration of the full isomiR-ome. 

## 4. Methods and Materials

*Setting and overview*—In this single tertiary center, proof-of-concept study, patients with suspected NAFLD from the outpatient liver clinic of Soroka University Medical Center (SUMC), were screened for eligibility using liver magnetic resonance spectrometry (MRS). Those with ≥5% (*wt*/*wt*) liver fat were considered as patients with established NAFLD [39] and were further evaluated with computed tomography (CT) for coronary calcium score. The coronary artery calcium (CAC) score and demographic parameters (sex, age, and race) were used to calculate the CAC score-based CV-risk percentile (CAC-CV%) [26,40]. The CAC-CV% was used as the reference standard for the assessment of CV risk. Using NGS as an unbiased approach to detect the circulating miRNA landscape, we quantified the miRNAs and their isomiRs for each patient and assessed their correlation with the CAC-CV% (Figure 1). The study protocol was approved by the local ethics committee of Soroka University Medical Center (0180-19-SOR). All experiments were performed in accordance with relevant guidelines and regulations.

*Patients*—Thirteen patients (*N* = 13), males and females, aged 46–79 years, with established (MRS-confirmed) NAFLD, who signed the informed consent form, were enrolled (Appendix A). Exclusion criteria included established CVD, viral or alcoholic hepatitis, any decompensated liver disease, severe debilitating disease, diagnosis of cancer 5 y prior to screening, chronic immunosuppressive or immune-modulating medications (including glucocorticoids), and inability to complete the imaging tests. Pregnant or breastfeeding women were also excluded. 

*Clinical and biochemical evaluation*—During enrolment, anthropometric and medical history, drug treatment, and CV-risk factors were reviewed. Fasting blood samples were drawn for complete blood count, biochemistry, glycated-hemoglobin (HbA1c), and miRNA profiling (see below). We determined metabolic syndrome criteria [41] and ‘10 years CV-risk’ using 3 calculators: The Framingham risk-score (FRAM) [42], the Systematic Coronary Risk Evaluation (age-based SCORE2 and SCORE2-OP) [43,44], and the American College of Cardiology/American Heart Association (ACC/AHA) CV-risk calculator [45].

*Coronary Calcium Scanning and magnetic resonance spectrometry*—Non-contrast cardiac-gated CT scans for coronary artery calcium were performed using standard clinical protocols. Coronary calcium score was quantified using the Agatston method [46]. Calculations of CAC score-based CV-risk percentiles (CAC-CV%) were based on the MESA cohort [26]. Measurements of liver fat were performed with a 3-Tesla magnetic resonance imaging (MRI) scanner, using MRS. The percentage of intrahepatic fat was displayed on the fat fraction images generated by the quantitative DIXON sequence. Five evenly spaced axial sections of the liver were selected, and five regions of interest (3–4 cm^2^) were drawn for each section. Care was taken to represent all the liver lobes in the image, while avoiding blood vessels and bile ducts. Mean fat percentage for each axial slice was determined, and then the mean percentage of fat of the entire liver was averaged [39].

*miRNA extraction, quality control and sequencing*—Plasma small-RNAs were extracted using miRNeasy Serum/Plasma Advanced Kit (Qiagen (Hilden, Germany), Cat-No.217204) according to the manufacturer’s protocol. *C. elegans* miRNA Cel-miR-39 (UCACCGGGUGUAAAUCAGCUUG) was added at 5 pg/sample as a technical control for miRNA recovery (=Spike-in) (Cel-miR-39 was custom-ordered from Thermo (Waltham, MA, USA), phosphate was added to the 5-prime of the sequence for compatibility to TaqMan assays). To ensure that circulating miRNA profile was not significantly contributed to by erythrocytes lysed during blood draw, we measured (using RT-qPCR) miR-451a and miR-23a-3p, representing miRNA contribution from red blood cells and plasma, respectively. (CT miR-23a-3p)–(CT miR-451a) < 7 indicated non-hemolytic sample [47], which were further sequenced by NGS to profile the miRNAs and their isomiRs. For sequencing, miRNA libraries were prepared using QIAseq™ miRNA Library Kit (Qiagen, Cat-no. 331505), following the manufacturer’s instructions. A 3′ adapter and then 5′ adapters were ligated to the extracted RNAs. cDNA was generated and cleaned using supplied magnetic beads, and libraries underwent amplification followed by an additional clean-up step. NGS was run on Illumina (San Diego, CA, USA) NextSeq instrument using NextSeq 500/550 High-Output Kit v2.5 (75 Cycles, Cat-no. 20024906). 

*Statistical analysis of clinical data*—Demographic, clinical, and biochemical parameters are shown for each patient (Appendix A). Continuous variables are presented as median (Q1, Q3), and categorical variables are displayed in counts and percentages. We used ‘R’ ver. 4.1.2 (rcorr command from Hmisc package) to execute Spearman correlations. As in an exploratory study, no formal calculation of sample size was conducted. 

*Bioinformatic analyses*—Fastq files were processed with Cutadapt [48] (cutadapt -*a* AACTGTAGGCACCATCAAT -*m* 17) to remove adapters, and assessed for quality using FastQC [49]. The quantification of small RNAs in the samples was performed with miRGE3 [50]. Herein, the term “miRNA” is used to refer to the full repertoire of the miRNAs, including the canonical sequence and its related variant isomiRs, while the term “isomiR” is used to refer to a specific sequence (canonical or variant). As an additional quality-control step, the rate of miRNA detection in the small RNA libraries was examined: subsets of common miRNAs that are present in at least 3 or 5 samples were identified, and for each library, the number of miRNAs present in these subsets was quantified. Outlier library was removed.

Using miRGE3 RPM file, the average and maximal abundance across all libraries were calculated for each miRNA. miRNAs with an average count > 5 RPM and a maximal count > 50 RPM were collected for further analysis. The non-parametric Spearman’s test was used on RPM-normalized counts to assess the correlation between miRNA abundance and CAC-CV%. For statistical significance, we used a permutation test on the results of correlation analyses between miRNAs and CAC-CV%, followed by multiple testing corrections, as follows: miRNAs with a Spearman’s r(ρ) ≥ 0.55 were further assessed by a ‘permutation test’, as previously described [51]. Briefly, the assignment of CAC-CV% to patients was shuffled and the correlation test was repeated 10,000 times. Then, for each miRNA, the number of times it received a better correlation in the shuffled data compared to the original data was recorded, and then divided by 10,000 to generate the permutation test’s *p*-value. These *p*-values were then corrected for multiple testing using Benjamini–Hochberg correction. A corrected *p*-value (*p*-adj) < 0.05 was considered significant. Since each miRNA represents all variants, including the canonical miRNA and multiple isomiRs [11,12], the filtration steps and correlation analysis were repeated and executed for isomiRs, using RPM-normalized isomiR counts as described above for the miRNAs. Receiver operating characteristic (ROC) curve analysis was performed on RPM values of miRNAs or isomiRs to evaluate their ability to discriminate patients with high/low risk using 75th percentile CAC-CV% as the cutoff. Discriminative miRNAs or isomiRs were those whose ROC-AUC low confidence interval (CI) value exceeded 0.5. 

To determine diseases and pathways associated with CAC-CV%-correlated miRNAs, an miRNA set enrichment analysis was performed with the online tool miEAA 2.0 [27] [Analysis type: Over-Representation Analysis (ORA), Categories: Pathways (KEGG) and Diseases (MNDR), *p*-value adjustment method: FDR (Benjamini–Hochberg method) adjustment, *p*-value adjustment scope: Adjust *p*-values for each category independently, Significance level: 0.05, and Minimum required hits per sub-category: 2]. 

Predicted targetomes of seed-changing, CAC-CV%-correlated isomiRs, and their respective canonical miRNA sequences were retrieved using the miRmut2go web-tool [28], with two algorithms: TargetScan and miRanda. Only genes that were predicted as targets by both programs were considered for the targetome. Overrepresented pathways (KEGG and Wikipaths) were identified using DAVID Bioinformatics Resources [29,52]. 

*Cholesterol Efflux Assay*—The cholesterol efflux assay protocol used was previously published [53]. Briefly, in a 24-well plate, Raw624.7 murine macrophages were incubated for 24 h with 3[H]-Cholesterol medium (Perkin-Elmer, Waltham, MA, USA) at 1 µCi/well, followed by incubation for 20 h in Cpt-cAMP buffer (Sigma Aldrich, St. Louis, MO, USA). Cells were then incubated with 2.8% Apo-B depleted plasma buffer in triplicates for 6 h and each plate included a blank condition with no plasma. Media and cell lysates were collected, 100 µL aliquots were mixed with 5 mL scintillation buffer (EcoLite(+), MP Biomedicals, Santa Ana, CA, USA) and were read for 10 min on a liquid scintillation analyzer (Tri-Carb 2100TR, Perkin-Elmer).

Apo-B depleted plasma buffer was prepared the previous day and was kept in −20 °C until used. Sixty microliters plasma were incubated with 24 µL 20% PEG-glycine buffer (Polyethylene Glycol, MW8000, Sigma Aldrich) at room temperature for 20 min before spinning the samples (30 min, 4 °C, 12,000 RCF). The supernatant was collected and spun again. The recovered Apo-B depleted plasma was mixed with HEPES medium containing 2 µg/mL Acyl-coenzyme A: cholesterol acyltransferase (ACAT) inhibitor (Sandoz, Sigma-Aldrich).

Prior to calculating cholesterol efflux, all triplicates were reviewed and any values whose difference from the mean triplicate value was 1.15 standard deviations or more were removed. This approach resulted in the removal of 1 replicate from 5 different samples. Cholesterol efflux values after this correction were strongly correlated with cholesterol efflux values before the correction (r = 0.87, *p* < 0.001).

Total percentage of efflux was calculated by dividing beta counts per minute (CPM) in the medium by the sum of CPM in the medium and CPM in the lysate and multiplying by 100%. Specific percent efflux was calculated by subtracting total efflux in the blank control from total efflux in each sample.

## 5. Conclusions

In conclusion, this proof-of-concept study demonstrates that considering the full landscape of circulating miRNAs, which includes the entire repertoire of isomiRs (i.e., the isomiR-ome), can uncover specific circulating miRNA species with a particularly high correlation with coronary risk, as assessed by CAC score. These are putative strong biomarkers that could be used to address the unmet need of better predicting high CV risk in patients with NAFLD.

## Figures and Tables

**Figure 1 ijms-25-00890-f001:**
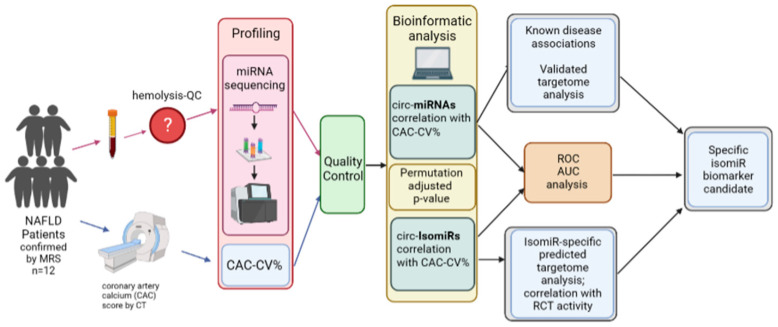
Overview of study design. NAFLD was defined as liver fat content ≥ 5% *wt*/*wt*, as measured by magnetic resonance spectrometry (MRS). Coronary atherosclerosis was assessed using CT-based coronary artery calcium (CAC) score, and a CAC-score-based CV-risk percentile (CAC-CV%) was calculated [26]. Blood was drawn to assess biochemical parameters, and miRNAs were extracted from plasma and sequenced by next-generation sequencing (NGS) after ensuring low levels of hemolysis. Bioinformatic analysis was performed to identify all circulating miRNAs (circ-miRNAs) and their isomiRs (circ-IsomiRs) which exhibit strong correlation with CAC-CV%, in order to identify putative biomarkers for increased CV risk in patients with NAFLD. The identified miRNAs and isomiRs were subjected to further statistical and functional analyses.

**Figure 2 ijms-25-00890-f002:**
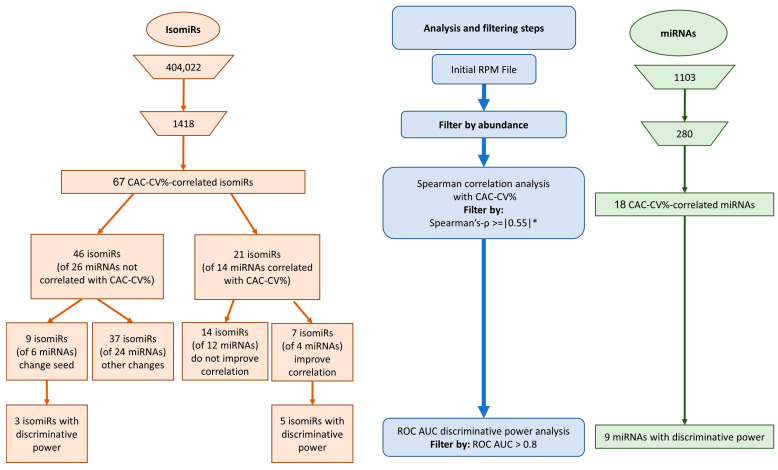
miRNA/isomiR analysis and filtering. Sequential filtration steps (in blue, middle) to search for miRNAs (in green, right) and isomiRs (in orange, left) that correlate with CV risk: filtration by abundance [(average-count > 5 RPM + maximal-count > 50 RPM) resulted in a decrease from 1103 to 280 miRNAs (right) and from 404,022 to 1418 isomiRs (left)]; filtration by correlation coefficient between miRNA/isomiR RPM counts and CV-risk percentile; * confirmation of the chosen miRNAs/isomiRs by permutation test on Spearman’s-ρ and FDR: *p*-adj < 0.05; ROC AUC discriminative power analysis. RPM—reads per million; FDR—false discovery rate.

**Figure 3 ijms-25-00890-f003:**
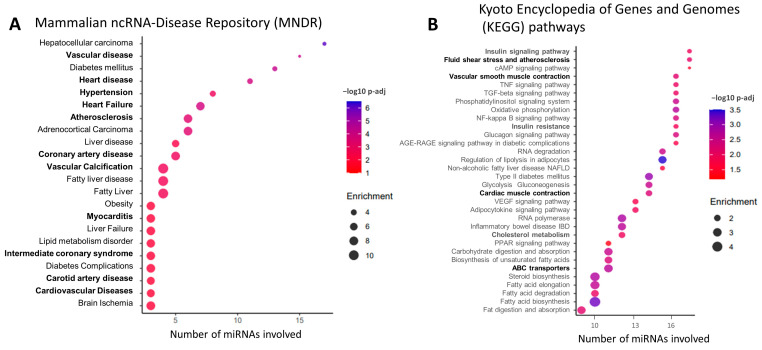
Associations of 18 CAC-CV%-correlated circulating miRNAs with diseases and KEGG pathways based on miEAA. Dot plots show over-represented (**A**) diseases and (**B**) KEGG pathways associated with the identified miRNAs (**A**) and their targets (**B**). The position of the dot on the x-axis corresponds to the number of miRNAs out of the 18 which are enriched in the term on the y-axis; the size and the color of the dot represent the enrichment score and the significance level, respectively. The terms are ordered based on the number of miRNAs on the x-axis. The full list of terms identified by miEAA is available in Appendix A.

**Figure 4 ijms-25-00890-f004:**
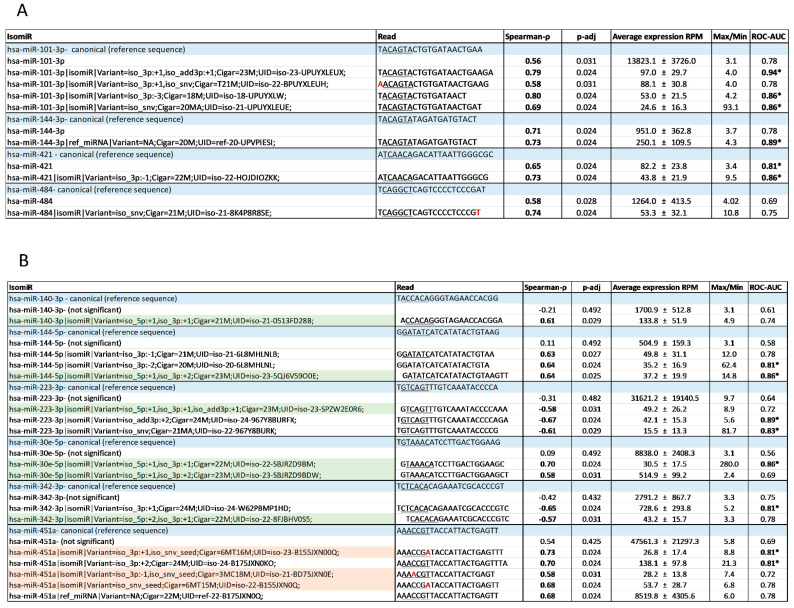
Circulating isomiRs with a stronger correlation with CAC-CV% compared to their respective miRNAs. (**A**) Four of the eighteen CAC-CV%-correlated circulating miRNAs had isomiRs with a stronger correlation compared to their corresponding (total) miRNA, as evident by higher Spearman’s coefficient of correlation (ρ), a lower *p*-adj, a larger dynamic range of abundance, and higher ROC-AUC. (**B**) Six miRNAs that did not significantly correlate with CAC-CV%, had at least one seed-changing isomiR that did correlate with CAC-CV%. isomiRs are grouped based on their respective miRNA. The canonical forms are listed for each miRNA (blue shade), the total miRNA set (canonical + isomiRs), and the identified isomiRs. The seed in each sequence is underlined. Variation in isomiR sequence compared to canonical sequence is shown in red. Seed-changing isomiRs are highlighted: green—a shift in the 5′ end, pink—nucleotide variation. miRNAs and isomiRs with Spearman’s coefficient of correlation (ρ) > |0.55| and ROC-AUC > 0.8 are marked in **bold**. Those with discriminative ROC-AUC (a lower confidence interval (CI) value > 0.5) are marked with *. A full list of CAC-CV% correlated isomiRs is provided in Appendix A.

**Figure 5 ijms-25-00890-f005:**
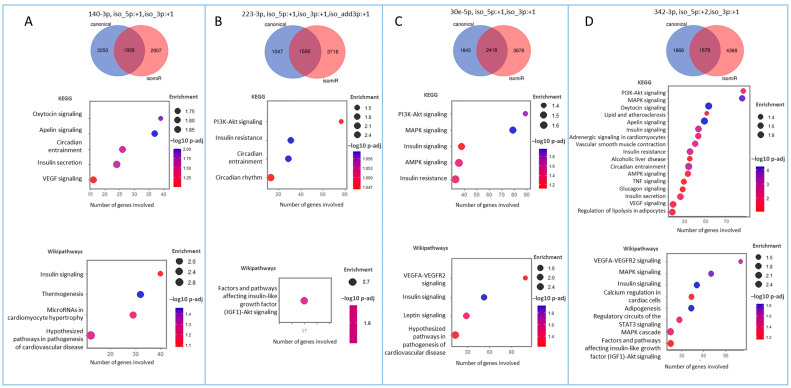
Pathway enrichment analysis for the predicted targets of seed-changing isomiRs. Targetomes of four of the seed-changing isomiRs, (**A**) 140-3p, (**B**) 223-3p, (**C**) 30e-5p, and (**D**) 342-5p, are enriched in cardiovascular disease-related pathways. Venn diagrams (top panels) show the overlap between the predicted targets of the canonical miRNA and the specific isomiR. Dot plots show over-represented KEGG pathways (middle panels) and Wikipathways (bottom panels) that are associated with the pathogenesis of NAFLD-related CVD and are significant (*p*-adj < 0.1) for the isomiR, but not for the canonical miRNA. The position of the dot on the x-axis corresponds to the number of target genes within the pathway on the y-axis; the size and the color of the dot represent the enrichment score and the significance level, respectively. The terms are ordered based on the number of genes on the x-axis. The full list of terms identified by DAVID is found in Appendix A.

**Figure 6 ijms-25-00890-f006:**
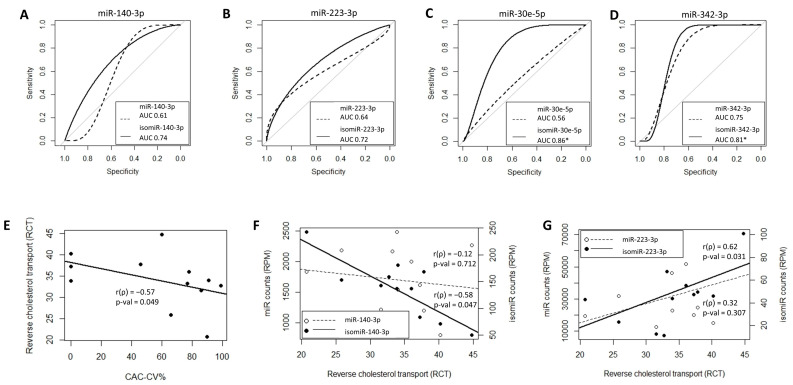
ROC-AUC analysis and association with reverse cholesterol transport (RCT). ROC curves and AUC values for four of the seed-changing isomiRs, (**A**) 140-3p, (**B**) 223-3p, (**C**) 30e-5p, and (**D**) 342-5p, show that isomiRs have a higher AUC compared to total miRNA. Two of them have a discriminative power, marked by *. The grey line in plots (**A**–**D)** represents the curve of random guess with AUC of 0.5. (**E**) RCT is inversely correlated with CAC-CV%. Two of the seed-changing CAC-CV%-correlated isomiRs, (**F**) 140-3p and (**G**) 223-3p, showed a significant inverse correlation and correlation, respectively, with RCT, while no correlation was observed with their respective total miRNA.

**Table 1 ijms-25-00890-t001:** Patients’ characteristics, biochemical parameters, CV-risk calculators’ score, and coronary artery calcium (CAC) score-based cardiovascular (CV) risk percentile.

	Median	Range (Min–Max)
Age	55	46–79
Sex (M/F)	9/4	
BMI (kg/m^2^)	30.8	26.4–42.7
Waist circumference (cm)	107	94–141
Dysglycemia (prediabetes or diabetes, yes/no)	7/6	
Hypertension (yes/no)	7/6	
Fasting plasma glucose (mg/dL)	104	80–222
Hb A1c (%)	5.8	4.6–9.4
Total Cholesterol (mg/dL)	182	94–254
Triglycerides (mg/dL)	145	82–210
Triglyceride/HDL	3.0	1.9–5.4
AST	29.0	18.0–59.0
ALT	33	11–82
Hepatic Fat (%) ^˄^	9.7	5.4–34.6
No. MS ^#^	3	1–4
Framingham CV-risk calculator ^§^	4.3	0.7–18.1
SCORE2/OP CV-risk calculator ^§^	4	2–13
ACC/AHA risk calculator ^§^	4.8	2.6–65.1
CAC-CV% ~^§^	66	0–99

^˄^ As measured by magnetic resonance spectrometry. ^#^ Metabolic syndrome criteria met by each participant. ^§^ 10-year risk (%) of cardiovascular disease. ~ coronary artery calcium (CAC)-based cardiovascular (CV) risk percentile was calculated using the coronary artery calcium score and demographic parameters (MESA calculator).

**Table 2 ijms-25-00890-t002:** Eighteen miRNAs that correlate significantly (*p*-adj < 0.05) with CV-risk percentile using Spearman’s correlation analysis.

miRNA	Average Expression RPM ^a^	Max/Min ^b^	Spearman-ρ	*p*-adj	Discriminative ROC-AUC
hsa-miR-185-5p	1766.7 ± 564.7	3.3	0.85	0.005	0.861
hsa-miR-20b-5p	210.5 ± 89.3	3.8	0.79	0.015	0.889
hsa-miR-548ad-5p	42.0 ± 15.2	3.9	0.74	0.018	0.889
hsa-miR-144-3p	951.0 ± 362.8	3.7	0.71	0.018	ND+
hsa-miR-15a-5p	655.3 ± 240.7	9.3	0.70	0.018	0.833
hsa-miR-106a-5p/17-5p	361.1 ± 99.2	2.5	0.69	0.018	0.889
hsa-miR-324-3p	43.0 ± 13.3	4.5	0.68	0.018	0.889
hsa-miR-106b-5p	88.5 ± 25.7	2.6	0.66	0.021	ND
hsa-miR-421	82.2 ± 23.8	3.4	0.65	0.020	0.806
hsa-miR-424-5p	81.2 ± 35.5	131.5	0.65	0.021	0.889
hsa-miR-20a-5p	1397.8 ± 334.2	2.5	0.63	0.023	0.889
hsa-miR-15b-5p	865.2 ± 233.0	2.7	0.59	0.028	ND
hsa-miR-484	1264.0 ± 413.5	4.0	0.58	0.028	ND
hsa-miR-25-3p	9153.4 ± 2534.3	2.3	0.58	0.028	ND
hsa-miR-101-3p	13,823.1 ± 3726.0	3.1	0.56	0.030	ND+
hsa-miR-1180-3p	65.2 ± 27.4	5.0	0.56	0.032	ND
hsa-miR-664a-5p	135.0 ± 43.7	4.0	−0.61	0.028	ND
hsa-miR-190b-5p	40.3 ± 17.0	55.9	−0.73	0.018	ND

^a^ The average RPM ± standard deviation is shown; ^b^ dynamic range of abundance: ratio between the highest abundance and lowest. Discriminative ROC-AUC are those with a lower confidence interval (CI) value > 0.5. ND—non-discriminative ROC-AUC; ND+—non-discriminative ROC-AUC for the corresponding (total) miRNA, but with at least one isomiR that has a discriminative ROC-AUC.

## Data Availability

Raw and processed miRNA-Seq data have been deposited in NCBI’s Gene Expression Omnibus (GEO) under GEO series record GSE199533.

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
