# Peer review of "Circulating isomiRs May Be Superior Biomarkers Compared to Their Corresponding miRNAs: A Pilot Biomarker Study of Using isomiR-Ome to Detect Coronary Calcium-Based Cardiovascular Risk in Patients with NAFLD"

_ijms, 2024, doi:10.3390/ijms25020890_

Round 1
Reviewer 1 Report
Comments and Suggestions for Authors
This manuscript is a study of the isomiR-ome for Detecting Excessive Cardiovascular Risk in Patients with NAFLD. The study propose a pipeline for exploring the circulating isomiR-ome as an approach to uncover novel and strong CVD biomarkers. It’s innovative and it’s a good job. Here, I still have the following questions.
1. The keywords in manuscript has not fully represent the main claim of this manuscript. It is recommended that the keywords in the manuscript be reconsidered.
2. The abstract is proposed that “NGS identified 404,011 different isomiRs”, but unique isomiR sequences is identified as 404,022 in Figure 2. Please check and explain.
3. Please clarify the age range of the patients in the manuscript. Lines 132 and line 265 are shown as “aged 45-80 years” and “range 47-79”, respectively. Supplemental Table S1-S2 shows the minimum age for 46 years old.
4. In Figure 2, how are isomiRs screened from 404,022 to 1418?
5. Authors must address some formatting errors throughout the manuscript. For example, line 229-231, 20 min or 20min, 30min or 30 min, 2ug/ml or 2 ug/ml.
Comments on the Quality of English LanguageMinor editing of English language required.
Author Response
Comments and Suggestions for Authors
This manuscript is a study of the isomiR-ome for Detecting Excessive Cardiovascular Risk in Patients with NAFLD. The study propose a pipeline for exploring the circulating isomiR-ome as an approach to uncover novel and strong CVD biomarkers. It’s innovative and it’s a good job. Here, I still have the following questions.
- The keywords in the manuscript has not fully represent the main claim of this manuscript. It is recommended that the keywords in the manuscript be reconsidered.
Response: Thank you for this suggestion. We have added the keywords cardiovascular risk, and isomiR (iso-microRNA), to the key words to better represent the scope of the manuscript.
- The abstract is proposed that “NGS identified 404,011 different isomiRs”, but unique isomiR sequences is identified as 404,022 in Figure 2. Please check and explain.
Response: Thanks for noticing this typo in the abstract. The correct number is 404,022, and this has been corrected.
- Please clarify the age range of the patients in the manuscript. Lines 132 and line 265 are shown as “aged 45-80 years” and “range 47-79”, respectively. Supplemental Table S1-S2 shows the minimum age for 46 years old.
Response: Thanks for noticing. The age range in this cohort is 46-79, with most (12 of 13) ranging 46 to 66. We corrected in all the relevant places, and changed the text accordingly.
- In Figure 2, how are isomiRs screened from 404,022 to 1418?
Response: This is the result of filtering by abundance – i.e., each isomiR, just like with the total miRNAs, had to be with average-count>5 RPM and maximal-count>50 RPM. We have clarified this in the legend for Figure 2.
- Authors must address some formatting errors throughout the manuscript. For example, line 229-231, 20 min or 20min, 30min or 30 min, 2ug/ml or 2 ug/ml.
Response: Thank you – this has now been corrected.
Reviewer 2 Report
Comments and Suggestions for Authors
In general, the data presented in the paper demonstrated the specificity and sensitivity of isomiRs as better biomarkers than miRNAs.
The author demonstrated a profound and conherent study procedure of miRNAs including sample collection, quality control, sequencing, Bioinformatics analysis, miRNAs quantification and assay validation.
The study of using isomiRs as biomarkers for CVD is limited due to their specificity and diversity. The paper represents a promising area in the biomarker field for CVD. However, more research is needed to fully understand their roles, validate their clinical utility, and overcome technological challenges.
Author Response
Comments and Suggestions for Authors
In general, the data presented in the paper demonstrated the specificity and sensitivity of isomiRs as better biomarkers than miRNAs.
The author demonstrated a profound and conherent study procedure of miRNAs including sample collection, quality control, sequencing, Bioinformatics analysis, miRNAs quantification and assay validation.
The study of using isomiRs as biomarkers for CVD is limited due to their specificity and diversity. The paper represents a promising area in the biomarker field for CVD. However, more research is needed to fully understand their roles, validate their clinical utility, and overcome technological challenges.
Response: Thank you very much for acknowledging the strengths of our study, while noticing its very clear limitations. We have devoted a section at the end of the Discussion to the study’s strengths and limitations. We have now added a sentence to further emphasize that significant more research is required to uncover the role and utility of circulating isomiRs in NAFLD-associated CVD risk assessment.
Reviewer 3 Report
Comments and Suggestions for Authors
The authors report the results of a pilot study (only 13 patients) which essentially took patients with non-alcoholic fatty liver, a known CVS event risk, and determined whether there was any significant correlation between a number of abundant isomiRs in plasma and the extent of coronary artery calcification. It was found that there were substantial correlations involving quite a few of the isomiRs. These are truly novel findings, and the authors should be complimented for undertaking this investigation.
However, there are lots of limitations here! These need to be admitted and briefly discussed.
(1) The authors use Framingham calculations of coronary event risk. The relevance of such formulae in the context of NAFLD is totally uncertain.
(2) The statistics were not formally corrected for multiple comparisons, nor was it possible to use multivariate analyses of data because of the small number of patients evaluated.
(3) Since the real comparison is with extent of coronary artery calcification, the title needs to be changed, and the limitations of utility of this measure should be admitted: while the average patient age was OK, calcium scoring has little utility in the elderly, and patients up to 79 years of age were included here.
(4) It is now appreciated (post ISCHEMIA and ORBITA studies) that the determinants of coronary event risk and of symptomatic ischaemia have much more to do with small coronary than large coronary disease, but the small coronary disease, like NAFLD, is mainly of inflammatory/"metabolic" origin. This should be discussed.
(5) The study was in no position to determine whether there is any likelihood that the correlations observed reflect a mechanistic association. No crime here: it just needs to be admitted and discussed.
Comments on the Quality of English LanguageOccasional spelling mistakes
Author Response
Comments and Suggestions for Authors
The authors report the results of a pilot study (only 13 patients) which essentially took patients with non-alcoholic fatty liver, a known CVS event risk, and determined whether there was any significant correlation between a number of abundant isomiRs in plasma and the extent of coronary artery calcification. It was found that there were substantial correlations involving quite a few of the isomiRs. These are truly novel findings, and the authors should be complimented for undertaking this investigation.
However, there are lots of limitations here! These need to be admitted and briefly discussed.
Response: Many thanks for this positive assessment of our study. We totally agree that this is a limited study, particularly given the small sample size, based on which we relate to it as proof-of-principle, and a pilot study. Moreover, we have refrained from putting much focus on the specific circulating miRNAs and isomiRs found, but rather presented the work as a pipeline that should inspire future studies. Finally, we elaborated on the strengths and limitations of the study, particularly in the last paragraph of the Discussion.
Thank you again for acknowledging the quality and contribution of this study to the field, and we hope to follow up on this pilot in the future.
- The authors use Framingham calculations of coronary event risk. The relevance of such formulae in the context of NAFLD is totally uncertain.
Response: We agree. The 3 clinically-accepted and frequently utilized calculators to estimate individual CV risk demonstrate that, in comparison to risk estimation by coronary calcium imaging, poorly perform in this patient population with NAFLD. This highlights the motivation for our study, as it demonstrates the need for novel biomarkers that would better indicate individual risk in NAFLD patients.
To clarify this point along your comment, we have added a sentence in the end of Results section “Patients and Clinical Parameters”.
- The statistics were not formally corrected for multiple comparisons, nor was it possible to use multivariate analyses of data because of the small number of patients evaluated.
Response: It is true that the small size of the cohort imposed statistical challenges. However, we did perform multiple testing correction (Benjamini-Hochberg test) on the miRNAs after a permutation test, that had a Spearman correlation coefficient ≥0.55. This is shown in Figure 2, presented in Methods (Bioinformatic analysis), and also explained in Discussion.
To nevertheless clarify this point, we added an “opening sentence” within the relevant section in Methods (“Bioinformatic analyses”): “For statistical significance, we used a permutation test on results of correlation analyses between miRNAs and CAC-CV%, followed by a multiple testing correction, as follows:”
- Since the real comparison is with extent of coronary artery calcification, the title needs to be changed, and the limitations of utility of this measure should be admitted: while the average patient age was OK, calcium scoring has little utility in the elderly, and patients up to 79 years of age were included here.
Response: i. We have modified the title, as required. It is now: “Circulating isomiRs May Be Superior Biomarkers to Their Corresponding Canonical miRNAs: A Pilot Biomarker
Study of the isomiR-ome to Detect Coronary Calcium-based Cardiovascular Risk in Patients with NAFLD”.
- As for the use of CAC in this cohort to estimate CV risk via coronary calcification – although we included one elderly woman (79y), the range of ages of the other 12 participants was 46-66 years. The decision not to exclude this patient was based on MESA (PMID: 16365194), which included participants 45 to 84 years of age. Nevertheless, we emphasized the information about the age range in this cohort in Results (section ‘Patients and Clinical Parameters’).
- It is now appreciated (post ISCHEMIA and ORBITA studies) that the determinants of coronary event risk and of symptomatic ischemia have much more to do with small coronary than large coronary disease, but the small coronary disease, like NAFLD, is mainly of inflammatory/"metabolic" origin. This should be discussed.
Response: Thank you for this comment. Indeed, the assessment of subclinical CVD in our study was limited only to its impact on large coronary vessels. We added a statement in the discussion section to address this deficiency, along with references of the two mentioned studies.
- The study was in no position to determine whether there is any likelihood that the correlations observed reflect a mechanistic association. No crime here: it just needs to be admitted and discussed.
Response: This is very true, and we relate to the reverse cholesterol transport studies as adding plausibility to our computational findings, not proving or reflecting mechanisms. Nevertheless, we have now added the following sentence at the end of the 3rd paragraph in the Discussion: “However, as our sample is small in this pilot study, the mechanism/s which connects miRNA with RCT, and other potential pathways related to atherosclerosis should be further investigated.”